# OpenReview forum: "LLM Sycophancy in Health and Climate Domains"
_ls2n.fr/CORIA-TALN/2026/Workshop/ARTS — ls2n CORIATALN 2026 Workshop ARTS Submission_

### Official Review · Reviewer_FLGT · 2026-04-27

**Mode De Presentation:** Oral

**Confience:**

Oui

**Decision:**

Accepté

**Relecture:**

Points forts :

- Un sujet actuel et important, avec un fort impact sociétal. Cela ouvre aussi des questions sur les performances de modèles ouverts (voir commentaire plus bas) et de modèles plus contraints en taille.

- L'introduction et l'état de l'art amènent bien le sujet, la problématique traitée ainsi que les questions de recherche soulevées.

- L'ajout d'une évaluation humaine ajoute à la robustesse des expérimentations

- La papier énonce clairement les limites de l'étude mais pose des questions de recherche pertinentes.

Points à améliorer :

- Manque de justification des modèles choisis :
	- Pourquoi être partie simplement sur des modèles commerciaux fermés et ne pas avoir évalué les compétences de modèles ouverts ? De même, les "petits modèles" sont de plus en plus utilisés, leur évaluation aurait apporté au papier.
	- Pourquoi ne pas s'être orienté vers des modèles par raisonnement ?

- Je ne comprends pas cette partie du protocole "Nous retenons 100 questions issues de l’édition 2021 et 100 questions issues de l’édition 2022 du TREC Health Misinformation Track" -> cela devrait donc faire 200 questions ? Mais je ne vois que 100 dans le tableau 1.

- Sauf erreur de ma part, la langue d'étude n'est pas précisée.

Autres remarques mineures :

- Attention aux termes "exhiber", "scalable"... qui font très anglais

- LLM-as-judge -> LLM-as-a-judge (qui me semble le plus utilisé dans le domaine)

- Résumé : pourquoi préciser le modèle précis pour GPT-5.1 mais pas pour Gemini (quel modèle ?)

- Je ne connais pas le thème, mais vérifier la manière dont sont faites les citations sous latex (voir notamment entre \cite \citet ...). La manière de les voir actuellement gêne la lecture (on attendrait par ex. des parenthèses avant/après la citation). Surtout dans l'introduction.

- Section 2 : "La complaisance (sycophancy) dans les grands modèles de langage (LLMs)" -> acronyme/traduction déjà fournis dans l'introduction

- Section 3.2 et 3.3 : je serais resté plus générique et je n'aurais pas parlé des modèles à ce niveau (car donné ensuite) - ie GPT5.1 et Gemini.

- ";. Le tableau 2 résume ces paramètres." -> supprimer le ;

- Attention aux séparateurs des valeurs dans les tableaux : en français, il faut des virgules et non des points.

- Pourquoi les tableaux 7 et 8 ne sont pas en % alors que c'est le cas pour 3 à 6 ?

- Claims -> faits

**Resume:**

Les auteurs proposent dans ce papier une étude sur la complaisance des LLM génératifs. Ils montrent que ces modèles peuvent adapter leurs réponses à la manière dont une question est posée plutôt qu'à des faits vérifiés, ce qui est particulièrement problématique dans des domaines sensibles comme la santé ou le climat. L'étude cherche à déterminer, d'une part, si les réponses courtes des modèles sont cohérentes avec leurs explications détaillées, et d'autre part si un autre modèle utilisé comme juge peut détecter cette complaisance de façon fiable par rapport à des évaluations humaines. Les auteurs ont testé un modèle (GPT-5.1) sur des jeux de données liés à la désinformation en santé et au climat, en utilisant un second modèle (Gemini) comme évaluateur automatique. Ils proposent également un protocole d'évaluation de la complaisance, qu'ils valident grâce à des annotations humaines indépendantes montrant un bon niveau d'accord entre évaluateurs.

---

### Official Review · Reviewer_6wBd · 2026-05-04

**Mode De Presentation:** Oral

**Confience:**

Oui

**Decision:**

Accepté

**Relecture:**

C'est un travail intéressant qui s'attaque à un comportement des GML encore peu étudié. Le protocole expérimental est solide et les résultats sont informatifs.
L'interprétation des résultats pourraient être facilitée en énonçant les règles de cette interprétation : c'est bien avoir un accord inter-annotateur élevé, en revanche c'est bien d'avoir un taux de complaisance bas.

Le travail de cette question sur les GML peut continuer et fournir d'autres résultats intéressants.



Néanmoins, la fiabilité de ces juges pour la détection de a complaisance ->
Néanmoins, la fiabilité de ces juges pour la détection de la complaisance

"métriques d’accord établies telles que le κ de Cohen et ." : phrase pas terminée

Dans la section 4.1, je compte 200 questions au total : 100 TREC + 100 Climate Fever. Si les questions à polarité inversée sont comptées, il faut l'indiquer clairement dans cette section.

"La graine aléatoire est fixée à 123 ;." : phrase pas terminée

**Resume:**

Ce travail s'intéresse aux réponses fournies par les GML. Les auteurs relèvent en particulier que les GML peuvent changer leurs réponses à des questions factuelles, selon la formulation de ces questions. Ce phénomène est appelé la complaisance des GML. Cela correspond donc à la problématique étudiée dans le travail. L'accent est mis sur l'impact du changement de la polarité dans les prompts.
Deux jeux de données sont utilisés : TREC Health Misinformation Track et Climate Fever. Pour chaque question, une version inversée est générée en conservant le contenu factuel identique, de sorte que toute variation de réponse est imputable à la seule formulation. Le modèle GPT-5.1 est évalué.

---

### Decision · Program_Chairs · 2026-05-07

Accept (Oral + Poster)